# Evaluation of Natural Pigments Production in Response to Various Stress Signals in Cell Lines of *Stenocereus queretaroensis*

**DOI:** 10.3390/plants11212948

**Published:** 2022-11-01

**Authors:** Jaime Abelardo Ceja-López, Javier Morales-Morales, Jorge Araujo-Sánchez, Wilma González Kantún, Angela Ku, María de Lourdes Miranda-Ham, Luis Carlos Rodriguez-Zapata, Enrique Castaño

**Affiliations:** 1Unidad de Bioquímica y Biología Molecular de Plantas, Centro de Investigación Científica de Yucatán, Calle 43, Número 130, Chuburná de Hidalgo, Mérida CP 97205, Yucatán, Mexico; 2Unidad de Biotecnología, Centro de Investigación Científica de Yucatán, Calle 43, Número 130, Chuburná de Hidalgo, Mérida CP 97205, Yucatán, Mexico

**Keywords:** betacyanins, betaxhantins, cactaceae, natural pigments, pitaya

## Abstract

*Stenocereus queretaroensis* (F.A.C. Weber ex Mathes.) Buxb is a cactus that has long been used as a source food in central and northern México. Its fruits, commonly called pitayas, biosynthesize high amounts of betalains. These molecules are water-soluble nitrogenous compounds; that compared to other pigments, such as anthocyanins or carotenoids, stand out for their physicochemical stability in industrial processes. Due to genetic and environmental factors involved in the biosynthesis and accumulation of secondary metabolites in plants, we tested different stress-inducing agents (elicitor, osmotic, salt, and temperature) to induce betalains accumulation in cell culture from fruits of *Stenocereus queretaroensis*. This work aimed to understand stress conditions that induce the metabolic pathways required for the accumulation of betalains. The results show how betacyanin concentration increases under high sugar conditions, thus affecting the expression of L-DOPA 4, 5 dioxygenase resulting in a strong dark red coloration. This suggests this enzyme is part of a rate-limiting step in betalain production. In addition, we found that betalains accumulation occurs under particular stress conditions. Cells that have a high level of betacyanins show better resistance to stress in the cell culture, as well as an overall different behavior including cell aggregation and alterations in nuclear size. Together the results shown here may provide new strategies to manipulate and mass produce the pigments from *Stenocereus queretaroensis* in cell culture.

## 1. Introduction

In general, plant pigments are secondary metabolites except for the chlorophylls which are photosynthetic pigments in all autotrophic organisms [1]. According to the solubility properties, the pigments can be classified into two groups: non-polar (liposoluble/fat-soluble) and polar (water-soluble). Examples include carotenoids (liposoluble) that have a long chain hydrocarbon derivate from isoprene, and anthocyanins (water-soluble) which have aromatics groups with ion-exchange charges [2]. Plants have impressive metabolic machinery with the possibility to synthesize complex molecules that nowadays are impossible to synthesize in efficient in vitro models. Thus, plants continue to be an essential source of raw materials for pharmaceutical, cosmetic, textile, and biotechnology industries. However, the metabolites yielded from plants are low, with around 1% based on dry weight [3], and in some cases, the metabolites are found in vital tissues, which complicates the survival of the plant. For that reason, it is essential to develop methods to obtain these metabolites. Some secondary metabolites are synthesized as a response to stress; therefore, cell cultures exposed to different conditions such as drought, salinity, UV radiation, etc., are capable of altering the metabolism and thus increases the number of secondary metabolites [4,5,6,7,8]. The cactaceae family has CAM metabolisms and grows in diverse regions of the world; in addition, they are emblematic plants of the desert areas in México since the pre-Columbian time [9,10,11]. They are in demand as a source of natural colorants for the food industry and can be an important alternative to the use of synthetic dyes [2,12,13,14,15,16]. *Stenocereus queretaroensis* (F.A.C. Weber ex Mathes.) Buxb. was recognized as a natural source of betalains [2,17,18,19]. This is an arborescent columnar cactus with a well-defined trunk that can reach up to eight meters in height. The shape of its fruits varies between ovoid or globose and are covered by a soft exocarp with thorny areoles which are detached in their phenological maturity during the spring; they weigh between 100 and 200 g and have very small pyriform seeds, its mesocarp is juicy and soft and range from a red to yellow and rarely greenish forms, depending on the variety. The mesocarp of this species is the vital tissue to obtain betalains [7,20]. Betalains pigments are a group of secondary metabolites in the Caryophyllales order [10]. They are water-soluble nitrogenous pigments derived from the amino acid tyrosine and can be chemically defined by the inclusion of betalamic acid as the central chromophore [21,22]. Betalains are subdivided into betacyanins (red) and betaxanthins (yellow). Compared to other pigments, such as anthocyanins and carotenoids, betalains have superior qualities, promising for their application in the industry; these include high water solubility, high stability, and color variability based on pH conditions [3,4,5,6,7], as well as the attractive characteristic of being odorless and tasteless [23]. Betalains are not only defined by these characteristics, but also by their bioactive qualities as anticarcinogens and antioxidants [24].

This work focused on the use of an elicitor (salicylic acid), osmostressors (PEG 6000 and sucrose), salts (NaCl), and temperature (45 °C) with the aim to test how various stress signals modulate or affect the biosynthesis of betalains and cell behavior *in vitro*. Additionally, we focused on exploring the pathways of betalains bioproduction. These pathways can lead to improved high-throughput models to produce pigments that can satisfy different industries and reduce the cost of additional sugars needed as a signal for the production of betalains in vitro.

## 2. Results

### 2.1. Pigments Quantification and Microscopic Characterization of sqY1 and sqR1 Lines of Stenocereus queretaroensis as a Study Model

The sqY1 suspension shows a distinct yellow color, which signals the presence of betaxanthin in the cells (Figure 1a). Characteristically, the sqR1 cell line shows a distinctive red color, which signals the presence of betacyanin (Figure 1b). Betalains (betaxanthins and betacyanins) were quantified throughout the culture from both of the sqY1 and sqR1 lines. The betaxanthin content remained stable, with a maximum at day 8 (18.41 µg g^−1^ FW). In the case of betacyanins, they present a small increase throught the culture time with an accumulation after 24 days from reseeding the culture (4.69 µg g^−1^ FW) (Figure 1a). In the sqR1 line, showed no dependence on the time after reseeding with small changes during the 24 days of the cultivation cycle. The betacyanin content showed a maximal amount after 8 days of reseeding (82.77 µg g^−1^ FW), whereas betaxanthins reached their maximum on day 16 (78.76 µg g^−1^ FW) (Figure 1b).

sqY1 cells were cultured in 3% sucrose, whereas sqR1 cells were cultured in 8% sucrose. They were harvested, fixed and stained with DAPI. [5]. Circular cells could be observed in both culture without nuclei alterations (Figure 1c: A,d: A) We detected a signal at 488 nm, in sqY1 cells only as fluorescent pigment (Figure 1c: B) that is absent in sqR1 expressing betacyanins (Figure 1d: B) This signal was distributed in the nuclear and cytoplasmic areas of the cell. The genes proposed for this study correspond to Tyrosine hydroxylase (TyrAa *ID: OP609705*), and 4,5 dioxygenase (DODA *ID: OP609705*), as key genes in the betalains biosynthetic pathway. The sequences were obtained from *Stenocereus queretaroensis* transcriptomic analysis (*accession number: GSE173230*). TyrAa gene transcripts showed a constant expression (Figure 1e). These results indicate a constant availability for the production of L-tyrosine. However, there is an increase in expression of the DODA gene which can result in an increase in the conversion of L-3,4-dihydroxyphenylalanine (L-DOPA), which is used by DODA to produce betalamic acid in sqR1 compared to sqY1 cell lines. The elongation factor gene sequence (EF-1α) was amplified as a control gene using the same amounts of cDNA since its expression remains constant during the tested growth conditions.

### 2.2. Signal Stress Effect of Exogenous SA Induction in sqY1 and sqR1 Cells Lines

Salicylic acid was applied to sqY1 and sqR1 cell lines on day 18 of the culture reseeding cycle. We harvest cells after 24, 48, and 72 h of treatment. In the sqY1 line, we did not observe any apparent changes during the first 48 h; however, after 72 h, cellular aggregation with a darker brown color occurred in all treatments of SA (Figure 2a), probably due to an increase in polyphenol content or cell death caused by the constant concentrations of SA. Since SA is involved in the expression of defense-related genes culminating in an increased production of lysis-inducing reactive oxygen species in cells. In contrast, the behavior of the sqR1 line showed no detectable color change under any of the study conditions or cell aggregation difference from the control experiment [6] (Figure 2b).

The content of betalains in the sqY1 cell line is affected by the different doses of SA. The betaxanthins increased slightly after 24 h followed by a decrease in concentration in all conditions dependent on the SA concentration. Betacyanin content increased after 24 h on the higher concentrations of SA. However, after that time, the content of betacyanin was similar to that of the control cells in all conditions (Figure 2c,e). In the treatments of sqR1 cell lines, the amount of betaxanthin showed significant changes after 48 and 72 h in SA exposure; on the other hand, betacyanins did not change significantly during all treatments, which is consistent with the observations of the cultures (Figure 2d,f).

### 2.3. Sucrose and Exogenous SA Application Involved in Color Variation

The sqR1 cell line was originally obtained by increasing sucrose in the medium of the sqY1 cell line. Here, we studied the effect of osmotic stress on betalains accumulation, sqY1 line cells were cultured in 3% sucrose basal medium and then changed to 3% sucrose or 8% sucrose plus 200 µM SA medium. The cells showed remarkable changes in pigmentation in the first 2 days of culture with the 8% sucrose treatment, and subsequent induction by the addition of SA generated a change in pigmentation from yellow to gray followed by an increase in cell aggregation (Figure 3a). The sqR1 line showed no change in pigmentation when cultured with the addition of 200 µM SA (Figure 3b). Quantification of total betalains showed that the change in sucrose from 3% to 8% induced the color change from yellow to red, thus betacyanin production increased concerning the control 3% sucrose, whereas the addition of 200 µM SA significantly decreased betalains production concerning the control (Figure 3c). On the other hand, the quantification of total betalains in the sqR1 line showed no significant changes with the addition of 200 µM SA (Figure 3c); however, treatment with 8% sucrose plus 200 µM SA in the sqY1 line induced the formation of gray cell clusters, and the change in sucrose from 3% to 8% promoted the staining of the cell clusters, compared to the control that remained yellow (Figure 3e). Cell aggregates in the sqR1 line showed no change in staining compared to the control (Figure 3f). Microscopic analysis of the sqY1 line under sucrose change from 3% to 8% and 8% sucrose plus 200 µM SA, staining with DAPI taken as nuclear reference, the nuclei were observed at 355 nm no different from the control (Figure 3g: A–C). Sucrose change from 3% to 8% showed a low autofluorescence signal whereas 8% treatment plus 200 µM SA showed no autofluorescence (Figure 3g: D–F). Treatment of sqR1 cells with 200 µM SA showed good definition, nuclei appeared to be intact (Figure 3h: A,B) but no betaxanthin fluorescence could be recorded at 488 nm (Figure 3g: C,D).

### 2.4. Exposure of Different NaCl Concentration to S. queretaroensis Cell Lines

The effect of salt stress induced by the addition of 100 mM was determined in the sqY1 and sqR1 cell lines. This treatment induced a change in cell pigmentation (from yellow to orange-red) after 48 h of exposure to salt stress (Figure 4a), whereas sqR1 cells retained the same color as untreated cells (Figure 4b). Quantification of total betalains showed significant differences in the production of betaxanthin and betacyanin, highlighting the increase in betacyanin content upon the addition of 100 mM NaCl in the sqY1 line (Figure 4c). The sqR1 line did not show significant changes concerning the control in the production of betaxanthin and betacyanin upon the addition of 100 mM NaCl (Figure 4d).

The addition of 100 mM NaCl in the sqY1 line promoted orange-red cell aggregates in contrast to the normal media as a control. (Figure 4e: A,B), whereas the sqR1 line remained without significant changes in cell pigmentation and aggregation concerning the control (Figure 4f: A,B). The cell nuclei in sqY1 showed no reduction in nuclear size concerning the control after 24, 48, and 72 h, which ranged in size by an average of 12.7 μm (Figure 4g: A–D); however, the sqR1 cell line showed changes in nuclear size that were not observed in the sqY1 cell line from 48 h under salt stress conditions, and the size of the nucleus under normal conditions ranged from 11.3 μm, and at 48 and 72 h the dimensions averaged 3.8 μm (Figure 4h: A–D). Regardless of nuclear size, cells continue to thrive under these conditions.

### 2.5. Osmotic Stress Effects on sqY1 and sqR1 Cell Lines

To induce osmotic stress to the cells, different concentrations of polyethylene glycol (3%, 4%, and 5%) were used in sqY1 presenting a light brown color (Figure 5a) and in sqR1 where the coloration of the cells remained unchanged (Figure 5b), regardless of the coloration, both lines continued their cell cycle. The quantification of total betalains in sqY1 at all PEG doses was downregulated compared to the control (Figure 5c), whereas in sqR1 there was no significant change in betalains production compared to the control (Figure 5d).

The formation of callus from cells suspension of the sqY1 line resulted in callus with brown coloration (Figure 5e: A–D); on the other hand, callus formed from the sqR1 line that developed normally and without changes in pigmentation concerning the control (Figure 5f: A–D). The localization of the nuclei with DAPI at 388 nm in the sqY1 line showed no nuclei affection (Figure 5g: A–D), and the autofluorescence at 488 nm was still perceived but the signal was affected (Figure 5g: D–G). In the same picture, the line cores are seen without damage in their shape (Figure 5h: A–D) whereas the sqR1 line does not show autofluorescence at 488 nm (Figure 5h: D–G).

### 2.6. Temperature Stress Effects on sqY1 and sqR1 Cell Lines

Exposure to 45 °C for 8 and 16 h in sqY1 and sqR1 cell lines did not induce pigmentation or cell aggregation (Figure 6a,b). However, total betalains levels increased after 8 h under heat stress in sqY1 (6c) and sqR1 (6d). Cell nuclei were not damaged by heat stress under this condition in sqY1 even after 16 h (Figure 6e). Cells and nuclei in sqR1 cell line appeared more organized and undamaged in their structure, possibly as a defense response to unfavorable conditions (Figure 6f).

### 2.7. Color Change Analyzer

The coordinates in CIELab space show that the color variation in the sqY1 line has a significant difference concerning the control line in the presence of all stress signals, among which the distribution of the points for each treatment in the chromatic circle of CIELab space showed distance from each other (Figure 7a); whereas, the sqR1 cell line showed no significant color variation concerning the control, and the coordinates in the chromatic circle showed that the distribution of the dots for each treatment was in red shades, with no variation to another red coloration concerning the control (Figure 7c). When plotting the coordinates of the CIELab space, we saw the distribution between the color difference for each treatment in the sqY1 and sqR1 lines, where the color variation in the sqY1 line showed greater variations in its coloration according to the CIELab space concerning the control (Figure 7b), while the sqR1 line the variation of coloration in the cells was minimal concerning the control in all study conditions (Figure 7d), which suggests that betacyanin may be related to the resistance to extreme conditions in *S. queretaroensis* in an open field.

## 3. Discussion

In recent years, betalains have received a renewed interest due to their physicochemical characteristics (high stability at wide ranges of temperature and pH) that can be of utility in many industries, as well as bioactive compounds for the prevention of some diseases [25,26,27,28]. Betalains are nitrogenous phenolic compounds derived from tyrosine and are exclusive to the order Caryophyllales. Betalains not only have the function of attracting pollinators and seed-dispersing animals, they also have a functional role in stress tolerance mechanisms [29].

*S. queretaroensis* fruits represent an important source of betalains, which accumulate in large quantities in their pulp. However, the fruits can be collected in limited numbers in only specific periods of time. Therefore, the uses of cell lines derived from fruits represent a viable alternative to study the regulatory mechanisms that control their synthesis [29,30,31]. The use of inductors of secondary metabolism is a long-used strategy to enhance the accumulation of certain compounds of interest. Among them, salicylic acid (SA) has been widely used to induce the biosynthesis of polyphenolic compounds. This inducer, together with methyl jasmonate, has been used to stimulate the production of flavonoids and polyphenols in cell suspensions, calluses, and tissue cultures of various plants [13]. The use of different doses of SA in *S. queretaroensis* cell suspensions seemed to favor the production of polyphenols over betalains accumulation. The use of sequencing tools, particularly RNA sequence analysis for de novo assemblies, have proven to be very useful in determining genes of importance for the synthesis of a large number of plant metabolites, including secondary metabolites derived from tyrosine such as alkaloids or other pigments such as anthocyanins [31,32,33,34] in model and non-model plants. The results of this approach are very useful for studying particularities in the synthesis of specialized metabolites derived from tyrosine; this being the case of betalains, in plants such as *S. queretaroensis,* whose fruits have the ability to produce a large number of betalains, giving them different colors, make it very attractive to deepen one’s knowledge of the synthesis of these pigments and metabolic engineering.

The TyrAa down-regulation in sqY1 cell line has been shown in *Arabidopsis thaliana* during cell growth [35]; however, in legumes, they did not show to be responsible for tyrosine biosynthesis [36]. However, in this regard there is little information from transcriptomic analysis from the order of Caryophyllales in general [37,38]. Abiotic factors, such as temperature, osmotic pressure, salinity, among others, represent a very important hindrance in relation to crop development and production in the field. It is known that temperature is an important factor in the cultivation of plant cells as it plays a key role in the stability of proteins [16]. Generally, temperatures used for cultivation are in the range of 25–35 °C. Temperature is closely related to the specific growth rate, the accumulation of intracellular reserve compounds (inactive biomass) and the rate of oxygen consumption. The regulation of osmotic pressure is essential in plant cell culture since it maintains water stability [17], and also greatly affects the production of bioactive compounds. Osmotic pressure is able to be regulated by using osmolytes that the cells can metabolize, such as sucrose or other sugars, or non-metabolizable osmolytes such as mannitol, sorbitol and polyethylene glycol (PEG). Several studies suggest that increased osmotic pressure results in slowing growth and increased intracellular accumulation of secondary metabolites in cultured cells [8,17].

A change in pigmentation (orange-reddish hue) could be observed in sqY1 cells in the presence of 100 mM NaCl, while this same treatment did not affect the sqR1 cells. When yellow cells were subjected to an increase in the concentration of sucrose from 3% to 8% sucrose, a reddish coloration appeared, which may be related to an increase in betacyanin, perhaps by redirecting the synthesis pathways to form the red pigments. The nuclear condensation on sqR1 cells reflects the sensitivity to salt stress in this cell line in comparison with the osmotic stress induced by the addition of PEG 6000 in the yellow cells. The changes by this particular stress in sqY1 resulted in remarkable changes in pigmentation and cell aggregation at all tested concentrations. sqR1 cells, which are cultivated in 8% sucrose, did show any significant change, suggesting that an additional osmotic disturbance do not further alter their betacyanin accumulation. Temperature stress did not induce any disturbance in pigmentation or cell development; therefore, it is suggested that betalains have a protective function at high temperatures, which is linked to the origin of the cell’s lines. The color change analysis using the CIELab system objectively demonstrated that the greatest color variation was present in the sqY1 cell line in all stress conditions compared to the control. The sqR1 line presented minimal variations in color change but without affecting the phenotype and cell proliferation in all stress conditions compared to the control as compared to the sqY1 line that is sensitive to all types of stress.

## 4. Materials and Methods

### 4.1. Biological Material, Propagation and Inoculation

Two *Stenocereus queretaroensis* (Weber) Buxbaum cell lines (sqY1 and sqR1) previously established from an immature pitaya fruit brought from Querétaro [18] were used. Both lines have been maintained in a Murashige and Skoog medium adapted for cacti, based on inorganic salts of MS (Sigma M5524), with a total calcium concentration of 4.4 µM and without nicotinic acid and glycine. It contained sucrose 3% for sqY1 cell line or 8% for sqR1 cell line(*w*/*v*), depending on the cell line, as well as 1-naphthaleneacetic acid (Sigma-Aldrich, St. Louis, MI, USA) (5.3 µM), 2,4-dichlorophenoxyacetic acid (Sigma-Aldrich) (4.5 µM) and kinetin (Sigma-Aldrich) (4.6 µM). Media pH was adjusted to 5.8 ± 0.02 and it was distributed at a rate of 40 mL in 250 mL Erlenmeyer flasks. Subsequently, these were sterilized in an autoclave at a pressure of 1 kg/cm^2^ and a temperature of 121 °C for 20 min. Each line produces mainly a particular class of betalains: the sqR1 line produces red-violet pigments (betacyanins) and the sqY1 line produces yellow pigments (betaxanthins).

Cell suspensions were propagated by subculturing every 15 days and maintained under the following conditions: temperature 25 ± 2 °C, light (4700 lumens) and continuous agitation (120 rpm). Each subculture was carried out by transferring 15–20 mL of the cell suspension to Erlenmeyer flasks with 40 mL of fresh culture medium.

In order to determine the growth cycle of the cell lines, an initial stock was prepared by pouring 15 flasks of suspensions into a one-liter Erlenmeyer flask (under sterile conditions). Once the cell stock was homogenized, 10 mL of the suspension, equivalent to 1.8 ± 0.2 g of fresh weight, were transferred to each of the 250 mL Erlenmeyer flasks with 40 mL of modified Murashige and Skoog culture medium for cacti. After the inoculation, the flasks were kept in the culture room under continuous light to 4700 lumens, a temperature of 25 ± 2 °C and 120 rpm, throughout the culture cycle.

### 4.2. SA induction

Cells from the sqY1 and sqR1 lines were collected on day 18 of their cell cycle, taking into account cell density. A 20 mM salicylic acid (SA) stock solution (Sigma-Aldrich) was prepared, dissolved in distilled water and sterilized by filtration at pH 5.7 (HI2211, HANNA^®^). The application was exogenous, adding 50 µM, 100 µM and 200 µM from the stock solution described above; at the same time, a blank was made by adding distilled water to the cell suspensions in the same volume as the elicitor, the treatments were performed in triplicate for each dose. Flasks with each treatment were placed in continuous light with orbital shaking at 120 rpm at a temperature of 25 ± 2 °C until harvesting after 24, 48 and 72 h of exposure to SA. Harvesting was performed using a vacuum pump (Model AR-1500 dry pump from Arsa^®^) and the filtered cells were stored in aluminum envelopes labeled with the corresponding treatment and stored at −80 °C until use.

### 4.3. Color Change Induction

From 14-day-old cells of the sqY1 and sqR1 lines growing culture room under a continuous light cycle, 10 mL was reseeded in a new modified MS culture medium for cacti. To induce color change, the sqY1 line was taken and the sucrose content of the culture medium was varied from 3% to 8% to mimic osmotic stress [1,2], in addition to an 8% sucrose treatment plus 200 µM SA from a 20 mM SA stock that was applied exogenously to the sqY1 and sqR1 lines. Flasks were placed in continuous light at a temperature of 25 ± 2 °C and 120 rpm orbital shaking, cells were grown by vacuum filtration after 9 days of exposure, stored in aluminum envelopes and brought to −80 °C until use, all treatments were performed in triplicate.

### 4.4. Salt Stress

Cells from the 14-day-old sqY1 and sqR1 lines were re-seeded in a new culture medium modified for cacti with the addition of 100 mM NaCl (Sigma-Aldrich) with 3% sucrose for the sqY1 line and 8% sucrose for the sqY1 line. The sqR1 line, then it was sterilized in an autoclave (Novatech, Lynchburg, VA, USA) at a pressure of 1 kg/cm^2^ and a temperature of 121 °C for 15 min. A total of 10 mL of cells was transferred to 3% for sqY1 cell line or 8% for sqR1 cell line with 100 mM NaCl, respectively. The flasks were kept under continuous light and at a temperature of 25 ± 2 °C with agitation at 120 rpm. After 48 h of exposure to the 100 mM NaCl treatment, cells were harvested by vacuum filtration, stored in aluminum envelopes and kept at −80 °C until use. The treatments were carried out in triplicate.

### 4.5. Induction Osmotic Stress

Osmotic stress was simulated using polyethylene glycol 6000 [3,4,5,6] (PEG 6000 MERCK) in sqY1 and sqR1 cell lines. In 250 mL Erlenmeyer flasks, 40 mL of cactus culture medium containing 3%, 4% and 5% PEG 6000 were added and sterilized in an autoclave (Novatech) at a pressure of 1 kg/cm^2^ and a temperature of 121 °C for 15 min; subsequently, using pipettes, 10 mL of cells from 14-day-old sqY1 and sqR1 lines were added due to their abundant cell development. The flasks were kept under continuous light and at a temperature of 25 ± 2 °C with agitation at 120 rpm. After 9 days of exposure to treatment, cells were harvested by vacuum filtration, stored in aluminum envelopes and kept at −80 °C until use. Treatments were performed in triplicate.

### 4.6. Temperature Stress

Cells from 10-day-old sqY1 and sqR1 lines were placed in an incubator (Tecnal TE-4200) and heated to 45 °C. During the induction of heat stress, the cells were maintained under normal growth conditions, agitation at 120 rpm and continuous light 4700 lumens Flasks were maintained there for 8 h at 120 rpm. Images were recorded before and after temperature exposure.

### 4.7. Pigment Extraction

For pigment extraction, two flasks were used for the sqY1 line and two for sqR1, to remove the culture medium a vacuum filter was used, 1g of fresh weight was used for each cell line, samples were placed in a cold mortar, liquid nitrogen was added, and they were then ground to a fine, uniform powder. The powdered sample was transferred to a test tube and ethanol was added in a 1:5 (*w*/*v*) ratio. With the help of the polytron, the sample was homogenized for 2 min; subsequently, it was centrifuged at 13,000 rpm for 20 min. The supernatant was transferred to a test tube, which was used to carry out all the determinations.

#### Quantitation of Betalains

Pigment content was determined by the spectrophotometric method, reported by Cai et al. [19], using the fresh extract obtained and using the following formula:
CB (mg/g) = A(PM)V(FD)/εLP
where CB is the content of betalains in mg/g of fresh weight, V is the volume, in mL, of the extract obtained, FD is the dilution factor, L is the length of the optical path in cm, and P is the weight of the sample in grams. In the case of red pigments (betacyanins), A is the absorbance measured at 538 nm; PM is the molecular mass of betanin (550 g/mol); ε, the molar absorptivity of betanin (60,000 L/mol × cm). For betaxanthins, A is the absorbance measured at 480 nm; MW is the molecular mass of indicaxanthin (308 g/mol); ε, the molar absorptivity of indicaxanthin (48,000 L/mol × cm). A spectrum analysis was carried out 300 to 800 nm in a DU 800 UV/Vis spectrophotometer Beckman Coulter to single out the peaks for the two wave lengths.

### 4.8. Microscopic Analysis

Fourteen-day-old sqY1 cells were cultured in 3% sucrose or 8% sucrose under above mentioned conditions. Color changes were documented on days 3, 6 and 9 of the culture cycle. Cells from day 9 were harvested and fixed with 4% formaldehyde (Sigma Aldrich) for 24 h at 4 °C. Cells were subsequently mounted in fluoroshield-DAPI (Sigma Aldrich) on glass slides and incubated for 24 h. Cell images were taken on a Axioplan Carl Zeiss microscope Axio Scope. A1 and confocal microscope (Olympus Fluoview FV500, Tokyo, Japan) using 388 nm for DAPI detection in blue auto fluorescence was detected at an excitation wavelength of 488 nm.

### 4.9. Callus Induction

Both lines were maintained in Murashige and Skoog medium adapted for cacti, based on inorganic MS salts (Sigma M5524, Darmstadt, Germany), with a total calcium concentration of 4.4 µM and without nicotinic acid and glycine. It contained 3% or 8% sucrose (*w*/*v*), depending on the cell line, as well as 1-naphthaleneacetic acid (Sigma-Aldrich) (5.3 µM), 2,4-dichlorophenoxyacetic acid (Sigma-Aldrich) (4.5 µM) and kinetin (Sigma-Aldrich) (4.6 µM). The pH of the medium was adjusted to 5.8 ± 0.02 and sterilized in an autoclave at a pressure of 1 kg/cm^2^ and a temperature of 121 °C for 20 min. It was distributed at a rate of 25 mL in plastic Petri dishes. Sterile 10 mL tips were used to collect cells in suspension from flasks of the sqY1 and sqR1 lines by placing a cluster of cells in the center of the Petri dish. The Petri dishes were then cellared and placed in continuous light at a temperature of 25 °C. Each line mainly produces a particular class of betalains: the sqR1 line produces red-violet pigments (betacyanins) and the sqY1 line produces yellow pigments (betaxanthins).

### 4.10. Color Change Analyzer

Images of the bottom of the flask were captured for each condition studied after day 9 under exposure to the stress signals. ImageJ software was used where each image per condition was analyzed separately by subtracting the background to eliminate impurities in the image. The values of L* luminosity, a* Red-green coordinates and b* Yellow-blue coordinates were calculated using the histogram tool where the values were subsequently plotted.

### 4.11. Statistical Analysis and Image Processing

All data is presented as means ± and standard deviations (SD) to obtain SD with three independent replicates for each treatment and analyzed as a completely randomized factorial design.

Group means were calculated and compared using one-way independent ANOVA analyses performed with Graphpad Prism 9.0.2 software.

For comparisons between treatments, Tukey’s honestly significant difference (HSD) tests differences in the means were considered statistically significant (*p* ≤ 0.05) was performed. The error bars are shown in charts with the represented error bars in the histogram. All figures were prepared with PowerPoint (Microsoft, Redmond, DA, USA), and ImageJ (https://imagej.nih.gov/ij/, accessed on 21 October 2022) for bans quantification and size of the nucleus.

### 4.12. RNA Extraction

Cells from each line were collected at specific stages of the growth cycle, based on betalains quantification. The sample collection was performed by vacuum filtration, where 0.1 g of cells were deposited in 20 μL cryotubes (CRYOKING^®^, Jinan, China) and finally stored at −80 °C until use. For total RNA extraction, the Chomczynski and Sacchi (1987) protocol was used, with slight modifications. Initially, 0.1 g per sample of cells stored at −80 °C from each line and day were used, individually homogenized in 1.5 mL Eppendorf tubes, in 750 μL of Trizol reagent (Life Technologies ™, Shanghai, China) with glass beads and vortexed. The tissue was manually homogenized with 300 μL of chloroform and incubated on ice for 3 min. Subsequently, the phases were separated by centrifuging at 12,000× *g* for 15 min at 4 °C, the aqueous phase was recovered in a new tube to which 500 μL of isopropanol was added, it was shaken and incubated at room temperature for 10 min, the mixture was centrifuged at 12,000× *g* for 15 min at 4 °C, carefully discarding the supernatant. The RNA pellet was washed in 1000 μL of 70% ethanol, vortexed, and centrifuged at 7500× *g* for 5 min. Three consecutive washes of the pellet were performed, adding 100 μL of 70% ethanol and centrifuging at 8000× *g* for 5 min at 4 °C each time. Finally, the tablet was resuspended in 30 μL of water with diethylpyrocarbonate (DEPC) and stored at −22 °C until use. The concentration and purity of the RNA obtained in each extraction method was confirmed by spectrophotometry (A260/A280 = 2.0, a ratio of 2.0 ensures a pure extraction) in a NanoDrop-1000 ND-1000 Spectrophotometer UV/VIS (ThermoScientific^®^, Waltham, MA, USA). The integrity and quality of the total RNA was verified by electrophoresis on a 1% agarose gel stained with SYBR^®^ Gold nucleic acid gel stain (Invitrogen^®^, Waltham, MA, USA).

### 4.13. Analysis of Expression of Genes TyrAa, DODA and EF1

The analysis of the expression of the *TyrAa* and *DODA* genes was carried out by RT-PCR. Specific primers such as Arogenate dehydrogenase (*TyrAa*, FW-ACAACTCACTGGGCATCTTC, RV-CTCTGGGAGCATAAGCAT) and 4, 5-extradiol dioxygenase (DODA, FW-CAAACTGGAAACTGGACATC, RV-TCCCAACTACGATG AATAAGCTC), in addition to the reference gene elongation factor alpha 1 (EF-1α, FW-CCAGAAACTACAGGTCCCAAC, RV-CCTCTCATGTTTCT CTTCAGCC), were used.

cDNA synthesis, the SuperScript ™ III First kit was used. Strand synthesis System for RT-PCR (Invitrogen). 1 µg of the total RNA previously extracted and treated with DNAses was taken and deposited in a mixture containing 1 µL of oligo (dT), 1 µL of dNTP mix (10 mM) brought to a volume of 10 μL with DEPC water and incubated at 65 °C for 5 min. For complementary strand synthesis, the cDNA synthesis mix was added, consisting of 2 μL of RT buffer (10X), 4 μL of MgCl2 (25 mM), 2 μL of DTT (0.1 M), 1 μL of RNAseOUTTM and 1 μL of SuperScript^®^ III RT (200 U/μL) and initially incubated at 50 °C for 50 min and finished by heating at 85 °C for 5 min. Finally, each reaction was treated with 1 μL of RNase H per tube and incubated at 37 °C for 20 min.

### 4.14. End-Point PCR

A total of 1 μg of cDNA was taken and incubated in a PCR mix containing 25 μL of DreamTaq Green PCR Master Mix (2X) and 10 μM of specific primers, brought to a total volume of 50 μL with nuclease-free water. The program consisted of 26 cycles that provided quantitative differences to be detected by PCR. The PCR products were resolved on 2% agarose gels and visualized in a transillumination system.

## 5. Conclusions

Results obtained from this work had been aimed to attain a better understanding of the regulatory expression of betalains synthesis, using a cell line of *S. queretaroensis* as an experimental model. The cell line sqY1 obtained from fruits of *S. queretaroensis* show a bright yellow color that under high sugar content changes to a dark red coloration due to increase betalain concentration resulting in the sqR1 cell line. The metabolic pathway for betalains production requires conversion of Tyrosine-by-Tyrosine hydroxylase to obtain L-DOPA followed by L-DOPA 4,5 dioxygenase to form betalamic acid. These two genes were explored in this analysis and show an increase in L-DOPA 4, 5 dioxygenase when sugar is added suggesting that this is a rate limiting step that may trigger betalains production. Obviously, other controlling mechanisms may be involved, including posttranscriptional modifications. Nevertheless, our results clearly point out that different conditions result in different behaviors from these two cell lines. With a lower sensitivity to salicylic acid, Sodium Chloride, temperature, and Osmotic stress in betalains containing cells. A deeper understanding of betalains production pathways should allow us to genetically engineer high-throughput models to produce pigments in sufficient amounts that can satisfy different industries. Here, the conditions presented not only show some possible mechanism that can be used to induce and affect betalains production, but also the stability of the culture under different conditions. The results may suggest that betalains accumulation under unfavorable conditions may represent not only an advantage for the plant in the field, but also provide us with strategies to manipulate their synthesis in more suitable experimental models, and hence, obtaining valuable basic knowledge of that in the future could substitute the use of some synthetic dyes.

## Figures and Tables

**Figure 1 plants-11-02948-f001:**
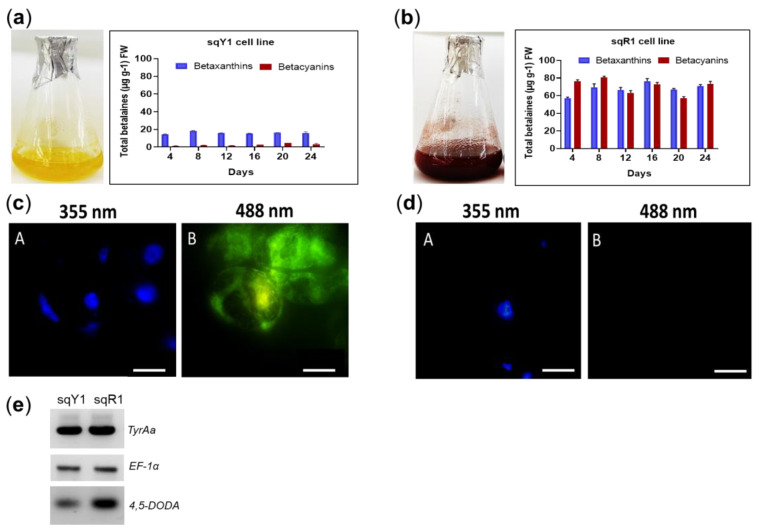
Pigments quantification and microscopic characterization of *S. queretaroensis* cell suspensions. (**a**) Pigments quantification on sqY1 line (yellow). (**b**) Pigments quantification on sqR1 line (red). The bar indicates a standard error of three independent replicates as described in material and methods. (**c**: A, B) Cells stained with DAPI at 355 nm and autofluorescence at 488 nm on sqY1 line. (**d**: A, B) Cells stained with DAPI at 355 nm and no autofluorescence at 488 nm on sqY1 line. Bar scale 10 µm with three independent replicates. (**e**) RT-PCR of TyrAa, DODA and *EF-1α* from sqY1 and sqR1 cells.

**Figure 2 plants-11-02948-f002:**
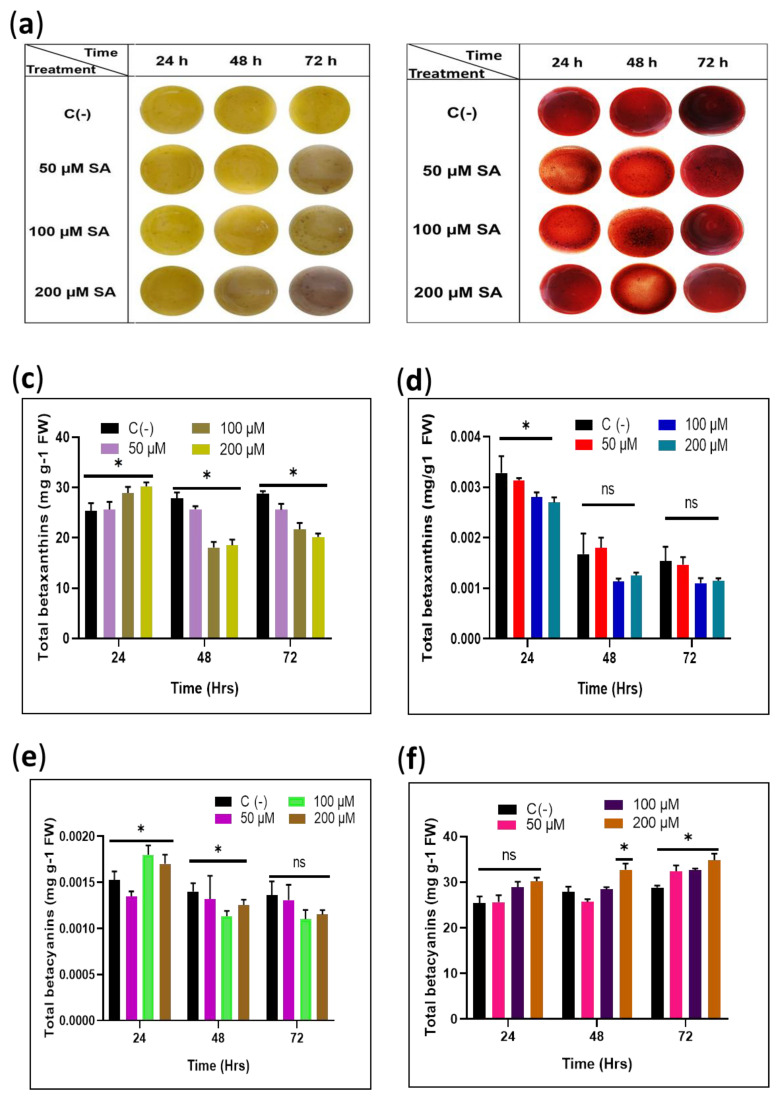
Evaluation of the effect of exogenous SA induction on total pigments in the sqY1 and sqR1 cells lines. (**a**) Culture cells after sedimentation from sqY1 cultivation under different time and treatment of SA. (**b**) Culture cells after sedimentation from sqR1 cultivation under different time and treatment of SA. (**c**,**e**) Quantification of betaxanthins and betacyanins for all treatments of sqY1. (**d**,**f**) Quantification of betaxanthins and betacyanins as stated in materials and methods for all treatments of sqR1. All experiments were obtained at a significance level of α = 0.05 with three independent replicates. Bar indicates standard error. (*) Indicate statistical significance at *p* < 0.05. (ns) Indicate no statistical significance.

**Figure 3 plants-11-02948-f003:**
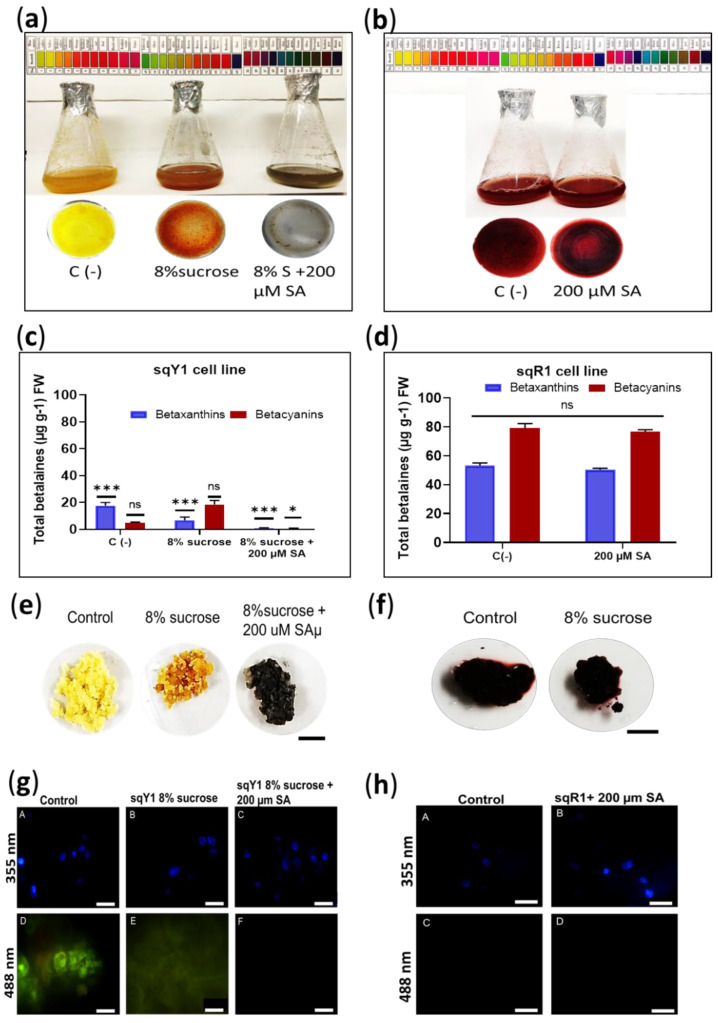
Effect of exogenous application of SA and variation of sucrose source in sqY1 and sqR1 lines. (**a**). Effect of exogenous SA application on sqY1 and (**b**) sqR1 cell lines. (**c**) Quantification of total betaxanthins and (**d**) total betacyanins in sqY1 line in response to exogenous SA application. (**e**) Cell cluster formation in the sqY1 line and (**f**) in the sqR1 line. Bar scale 1 cm. (**g**) Epifluorescence microscopy of the subcellular structure under the study conditions in the sqY1 line (**h**) and the sqR1 line. Bar scale 10 µm. All experiments were obtained at a significance level with three independents replicates. (*) Indicate statistical significance at *p* < 0.05, (***) Indicate statistical significance at *p* < 0.001. (ns) Indicate no statistical significance, as described in Materials and Methods.

**Figure 4 plants-11-02948-f004:**
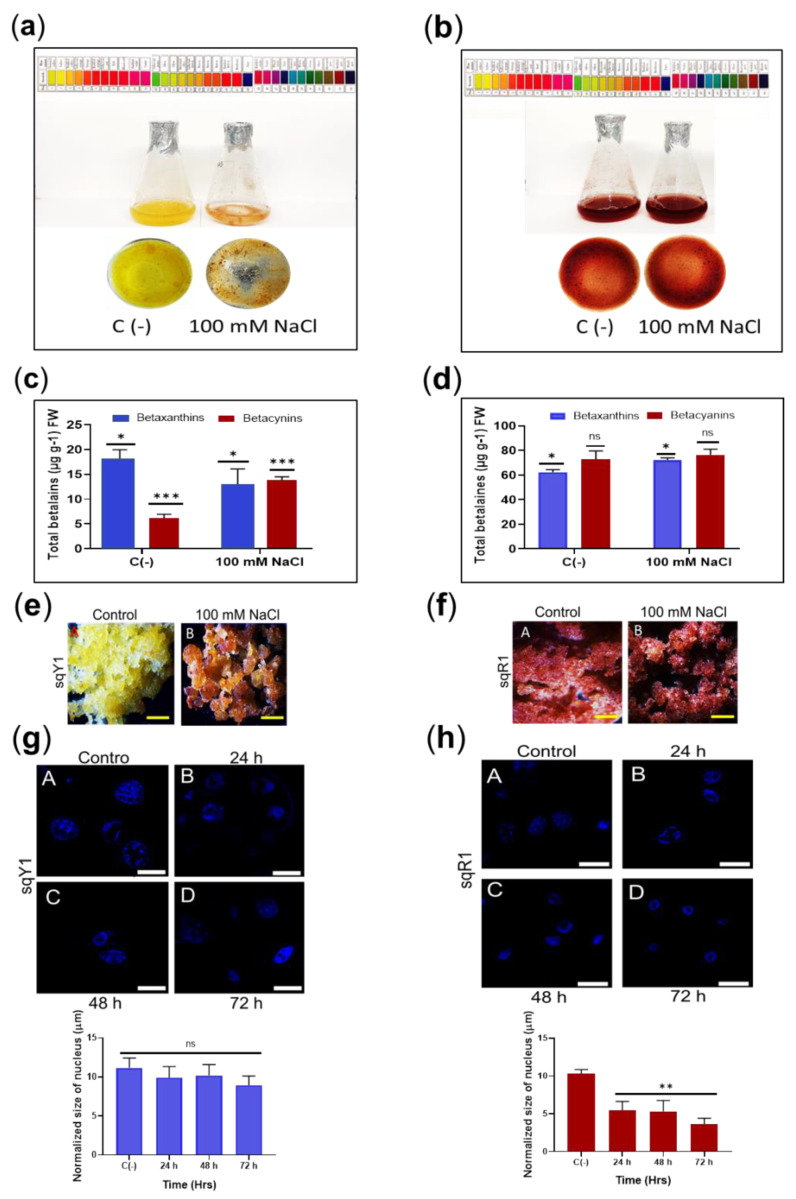
Salt stress in sqY1 and sqR1 cell lines. (**a**) effect of NaCl application in sqY1 line and (**b**) in sqR1 line (**c**) total quantification of betaxanthin and betacyanin under 100 mM NaCl conditions in sqY1 line and (**d**) in sqR1 line. (**e**) Cluster formation in line sqY1. Additionally, (**f**) in line sqR1. Bar scale 1 cm. (**g**) Nucleus measurements in line sqY1 and (**h**) in line sqR1. Bar scale 10 µm. All experiments were obtained at a significance level with three independents replicates. Bars indicates standard error. (*) Indicate statistical significance at *p* < 0.05, (**) Indicate statistical significance at *p* < 0.01, (***) Indicate statistical significance at *p* < 0.001. (ns) Indicate no statistical significance, as described in Materials and Methods.

**Figure 5 plants-11-02948-f005:**
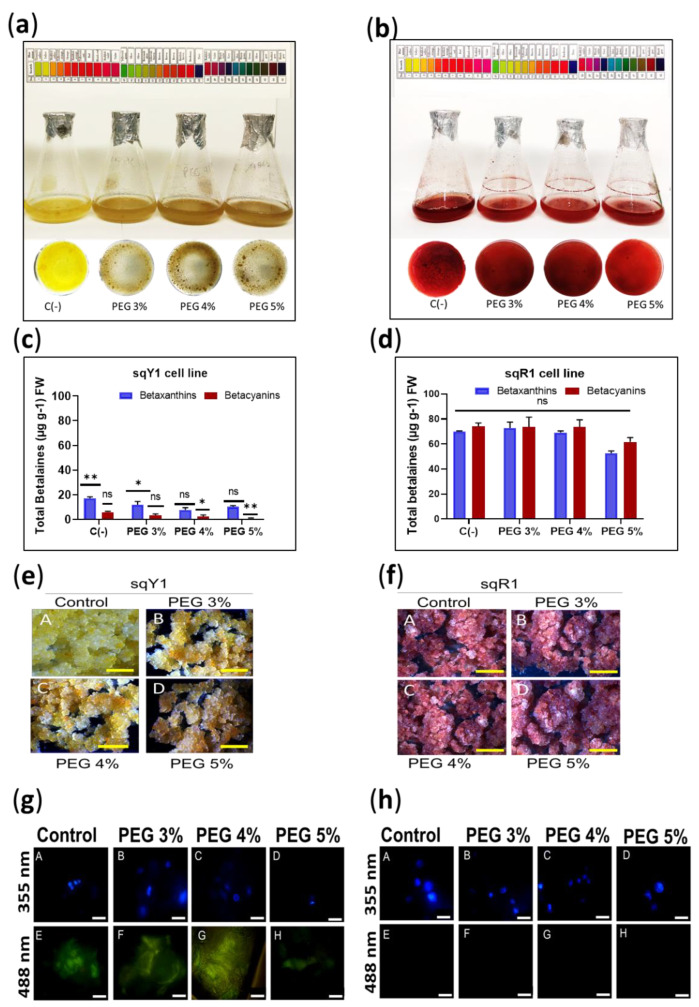
Induction of osmotic stress in sqY1 and sqR1 lines. (**a**) Effect of osmotic stress induced by the addition of PEG 6000 on sqY1 and (**b**) sqR1 cell lines. (**c**) Quantification of total betalains in sqY1 and (**d**) sqR1 lines. All experiments were obtained at a significance level whit three independents replicates. Bars indicate standard error. (*) Indicate statistical significance at *p* < 0.05, (**) Indicate statistical significance at *p* < 0.01. (ns) Indicate no statistical significance.Bars. (**e**) Variation of cell aggregate formation at different doses of PEG 6000 in the sqY1 line and (**f**) in the sqR1 line. Bar scale 1cm. (**g**: A–D) Epifluorescence microscopy, nuclei stained with DAPI and viewed at 355 nm in the sqY1 line (**g**: E–H) autofluorescence signal at 488 nm in the sqY1 line. (**h**: A–D) Visualization of nuclei at 355 nm stained with DAPI in the sqR1 cell line. (**h**: E–H) No autofluorescence signal was detected at 488 nm in the sqR1 cell line. Bar scale 10 μm.

**Figure 6 plants-11-02948-f006:**
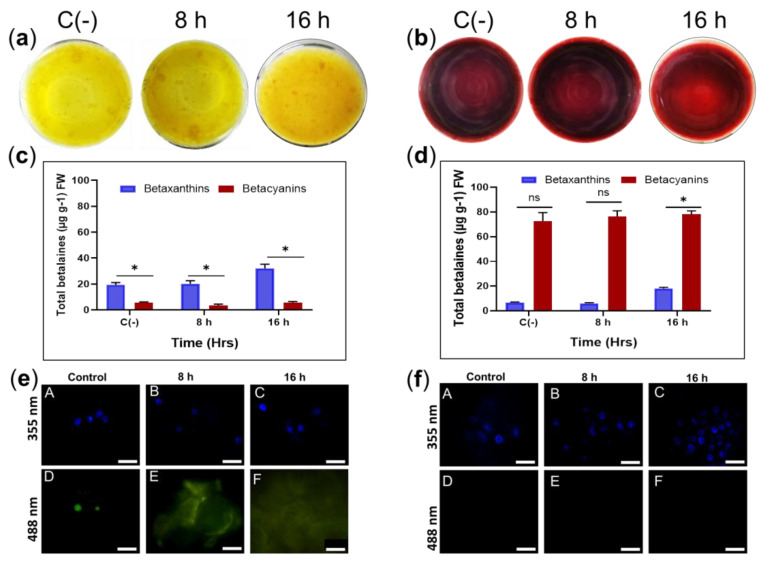
Effect of temperature on sqY1 and sqR1 cell lines. (**a**). Behaviors of sqY1 and (**b**) sqR1 cell lines under thermal stress at 45 °C for 8 and 16 h. (**c**) Quantification of total betalains in the sqY1 line and (**d**) in the sqR1 line. All experiments were obtained at a significance level whit three independents replicates. Bars indicate standard error. (*) Indicate statistical significance at *p* < 0.05. (ns) Indicate no statistical significance. Bar indicates standard error (**e**: A–C). Localization of nuclei stained with DAPI at 388 nm and in sqY1 and (**f**: A–C) in the sqR1 line. (**e**: D, E, F) Autofluorescence at 488 nm in the sqY1 line. (**f**: D–F) Absence of autofluorescence at 488 in the sqR1 line. Bar scale 10 μm.

**Figure 7 plants-11-02948-f007:**
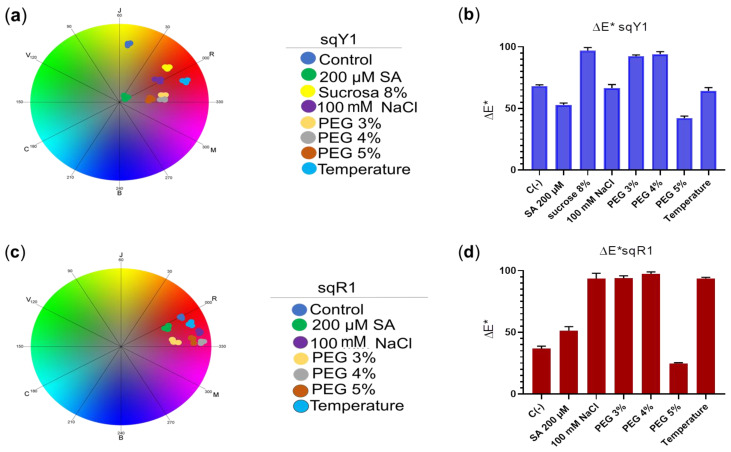
Analysis of color change in sqY1 and sqR1 lines under different types of stress. (**a**) Coordinates in the chromatic circle of CIELab space for the sqY1 line and (**c**) for sqR1, circles correspond to three replicates per treatment. (**b**,**d**) Bar graph shows the average color difference for each treatment in sqY1 and sqY2, respectably. Bars indicate standard deviation with three independents replicates.

## Data Availability

All Data is presented in the manuscript.

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
