# Peer review of "Evaluation of Natural Pigments Production in Response to Various Stress Signals in Cell Lines of Stenocereus queretaroensis"

_plants, 2022, doi:10.3390/plants11212948_

Round 1
Reviewer 1 Report
This manuscript entitled “Natural pigments production in response to various stress signals in cell lines of Stenocereus queretaroensis (F.A.C.Weber ex Mathes.) Buxb" is in journal topic and could be of interest for plant scientific community although some other similar studies had been published. The topic is relevant but not original and the paper is very descriptive with a limited discussion according to presented results; presented conclusions are ok. Moreover, the manuscript is well-written and can be easily read but style might be checked.
Abstract:
Ok
Introduction
Ok
L63 : osmostressor : uppercase
Material method
Cell suspensions were maintained under light conditions. However, no information characterize this parameter (intensity, light spectrum etc…)
L313 : precise which cells are cultivated with each sucrose solution
L337 and 352, 363 : needless to say that the author used a micropipette and tips..
L 381 : why authors used 45°C ? Are cells cultivated with light ?
L 396 : Unit problem : mg ou g ? !!!
L417 : « were used » must be deleted
L435 : defined k* a* and b*
Results
Results are consistent and well described. However, the sizes of figures are too small and make difficult the observation of each part of each figure. Moreover the yellow colors is not the best color to be used (difficulties to see the histogram bars). We can not see some legends when paper is print. We do not see the bar scale for microscopy pictures. Optical microscopy pictures could be added.
L 80 : unit error 82.77 µg g-1 FW and not FP
L113-115 page 3 : last sentence is wrong according to figures!
L 132 : from yellow to gray and not yellow to red.
Pag 5 : the figure is particularly illegible: too small!!
Page 5 fig 4 : g and h : too small
Page 9 : paragraph 2.7 : check the police
Discussion
L 260 : Betalains: uppercase to add
Author Response
We like to thank the reviewer for highlighting these problems, we have addressed all of your comments so it would be clear in all instances.
L63 : osmostressor : uppercase Fixed
Material method
Cell suspensions were maintained under light conditions. However, no information characterize this parameter (intensity, light spectrum etc…)
L313 : precise which cells are cultivated with each sucrose solution Thank you for your comment, we have described the sucrose percentage for each cell line.
L337 and 352, 363 : needless to say that the author used a micropipette and tips.. Thank you for your observation, we omitted that part.
L 381 : why authors used 45°C ? Are cells cultivated with light? We consider that 45 we get thermal stress, so let's test this temperature in our cell lines, we have detailed it
We consider a temperature of 45 to be adequate to induce heat stress. During heat induction the cells were maintained under normal growth conditions, agitation at 124 rpm, and continuous light at 4700 lumens in an incubator with the following characteristics
L 396 : Unit problem : mg ou g ? !!! Thank you for your observation, we have corrected mg.
L417 : « were used » must be deleted Fixed
L435 : defined L* a* and b* Thank you for your comment, we have defined L* a* and b*
Results
Results are consistent and well described. However, the sizes of figures are too small and make difficult the observation of each part of each figure. Moreover the yellow colors is not the best color to be used (difficulties to see the histogram bars). We can not see some legends when paper is print. We do not see the bar scale for microscopy pictures. Optical microscopy pictures could be added.
We appreciate all your observations; we have treated each of them with caution.
L 80 : unit error 82.77 µg g-1 FW and not FP: Fixed
L113-115 page 3 : last sentence is wrong according to figures! Thank you for your observation, we have rewritten
L 132 : from yellow to gray and not yellow to red.Fixed
Pag 5 : the figure is particularly illegible: too small!!: Thanks for the comment we agree an have increase the size
Page 5 fig 4 : g and h : too small Fixed
Page 9 : paragraph 2.7 : check the police
Discussion
L 260 : Betalains: uppercase to add Fixed
Reviewer 2 Report
Authors have investigated various stress-signal associated metabolic pathways with an aim to understand those which enhance production of betacyanin pigments. The presented work is worthy of publication. However, I have a few concerns, as under:
1. In my opinion, the title doesn’t depict the work presented in the manuscript. It should have been Evaluation of various stress signals…… in cell lines …
2. No authority in title. Only at the first instance in abstract and in the main text.
3. Abstract needs revision; hardly any results are there. It is all introduction and what is presented.
4. Line 26, comma after In general
5. Line 26-31, seem out of place, better merge in next para
6. Line 33; replace ‘reproduce’ with a better option
7. Line 42; It should be Cactaceae.
8. Line 46; it is not species!
9. Line 63; better mention the details of stress treatments in parenthesis.
10. Line 254; discussion, join first two para!
11. Line 313; why so much difference in sucrose levels!
12. Line 364; what was the rationale for the chosen conc of NaCLl
13. Line 372; Provide stress levels as Osmotic pressure, and not as % PEG!
14. Line 381: was performed?
15. Line 386: how much sample was taken for extraction?
16. Line 395; Provide details of Cai et al.
17. Sub section 4.8; any citations?
18. Line 417: were used?
19. Line 432: how images were taken?
20. Line 446: conclusion: should be revised and given in 3-4 sentences as standalone findings of the work.
Author Response
In my opinion, the title doesn’t depict the work presented in the manuscript. It should have been Evaluation of various stress signals…… in cell lines … Thank you for your comment, we agree and have changed the title
- No authority in title. Only at the first instance in abstract and in the main text. Thanks, we agree and have fixed it acordingly.
- Abstract needs revision; hardly any results are there. It is all introduction and what is presented. Thank you for your observation, we agree and have rethought the abstract
- Line 26, comma after in general Fixed
- Line 26-31, seem out of place, better merge in next para Fixed
- Line 33; replace ‘reproduce’ with a better option Fixed
- Line 42; It should be Cactaceae.Fixed
- Line 46; it is not species! Thank you for your observation, we have corrected it
- Line 63; better mention the details of stress treatments in parenthesis. Fixed
- Line 254; discussion, join first two para! Fixed
- Line 313; why so much difference in sucrose levels! Thank you for your question, we have better described this part
- Line 364; what was the rationale for the chosen conc of NaCLlWe observed through previous tests that 100 mM NaCl caused changes in pigmentation, therefore we decided to study the behavior of our cell lines with 100 mM NaCl
- Line 381: was performed? Fixed
- Line 386: how much sample was taken for extraction? Thank you for your question, we have defined how much sample was taken for extraction
- Line 395; Provide details of Cai et al. Fixed
- Sub section 4.8; any citations? Fixed
- Line 417: were used? Thank you for your comment, we have corrected it
- Line 432: how images were taken? Thank you for your question, we have defined how images were taken
- Line 446: conclusion: should be revised and given in 3-4 sentences as standalone findings of the work. Thank you for your comment, we have rewritten the conclusion
Reviewer 3 Report
The Authors did not clearly set out the broad purpose of the work. The described fact of the impact of some stressors on pigment synthesis presented in the paper is known from the literature. The choice of this particular plant has also not been adequately justified from the point of view of pigment production. I therefore find no novelty in the research performed. Furthermore, the authors did not presented statistical analyses in all Figures, but made deductions on the basis of the results (fig 1) or did not give precise information about the statistical analyses in the figures' descriptions. The methods used are not fully suitable for quantitative research. It is not only the two dyes that are measured by the chosen methods. The shown aggregation also influences the results (Figure 5). In any case, there is no validation of the methods used in the paper in terms of matrix and aggregation. The paper should have used more modern methods as shown by the state of the art - HPLC-MS, for example. Many of the figures need to be improved, there are not clear elements in them, so they should not be shown, or they should be bigger. The conclusions do not correspond to the research carried out. Rather, they specify what should be done, which makes the work unattractive to potential readers. Moreover, the bibliography is incorrectly built. The work in this version is not suitable for publication.
Author Response
We like thank the reviewer for his comments, as they have helped to improve this manuscript. This particular plant is produced commercially for several decades in the northern part of Mexico. However, its distribution it’s locally, similar to many fruits that are produced in a particular region, their shelf life is too short for export and therefore remains for local trade. Pitaya the fruit produced by this plant has this short distribution. However, the compounds found in the fruit in particular the Betalains that produce the intense red color can be used as stated in the manuscript for the food industry, pharmaceutical, and cosmetic industries. This compounds lack added flavors and lack an immunogenic response therefore qualified for industrial production. That is the reason for in vitro cell production. To have a culture that can be expanded and provide a realistic source of this compound. However, cell cultivation of the fruits provides a yellow color and the transition to red pigments can be achieved with the addition of large amounts of sugar. Our hypothesis is that a stress mechanism is what is providing the expression of enzymes required for the Betalain production. All of these parameters are needed to be tested and define what would be cost-effective and what to expect from such conditions. We definitely agree that there are state-of-the-art methods that would provide better information like HPLC-MS, it would be truly great if we could have the funds to do it. Sadly the economic situation here has not been kind and we lack the collaborations to do so at this point. However, we did find two of the genes from the metabolic route conversion and carried out RT-PCR. We tested for Tyrosine hydroxylase (TyrAa), and 4,5 dioxygenase (DODA), as key genes in the betalain biosynthetic pathway. We made sure that RT-PCR conditions were set on the logarithmic phase of the amplification by doing several PCRs at different cycles (Sadly we do not have a Realtime PCR machine at this time). We have added this information to the new figure 1. The spectrometry method we use in this manuscript has been used previously for this purpose (Cai et al 1998), we carry a scan of the spectra from 300 to 800 nm in a DU 800 uv/vis spectrophotometer Beckman Coulter. Since this species lacks carotenoids that could also provide a signal at 538nm which for Caryophyllales leads to betalains as a metabolite that can be measure at this particular wave rage. The betanin from Sigma-Aldrich is used as a control used to measure betalain content, similar we use 480nm which is indicated for betaxanins, at this wavelength, these are the main compounds that can be detected. This information has now been added, as you are correct that a simple absorption without a previous scan could lead to the possibility of other substances affecting the measurements. Also, the conclusions have been focused on the research that was done as suggested.
We also agree and have fixed each of the figures so that they could be properly seen.
Finally, we have checked and formatted the references according to the publisher using https://www.mendeley.com/reference-management/reference-manager
Round 2
Reviewer 2 Report
Authors have responded to the concerns/queries of the reviewers. The manuscript has been considerably revised. It is worth publication now.
Author Response
We like to thank the reviewer for his comments, we include here the new abstract as suggested with the added results from this study
Abstract: Stenocereus queretaroensis (F.A.C.Weber ex Mathes.) Buxb. Is a cactus that has long been used as a source food in central and northern México. Its fruits, commonly called pitayas, biosynthesize high amounts of betalains. These molecules are water-soluble nitrogenous compounds; that compared to other pigments, such as anthocyanins or carotenoids; stand out for their physicochemical stability in industrial processes. Due to genetic and environmental factors involved in the biosynthesis and accumulation of secondary metabolites in plants, we tested different stress-inducing agents (elicitor, osmotic, salt, and temperature) to induce betalains accumulation in cell culture from fruits of Stenocereus queretaroensis. This work aimed to understand stress conditions that induce the metabolic pathways required for the accumulation of betalains. The results show how betacyanin concentration is increased under high sugar conditions, thus affecting the expression of L-DOPA 4, 5 dioxygenase resulting in a strong dark red coloration. This suggests this enzyme is part of a rate-limiting step in betalain production. In addition, we found that betalains accumulation occurs under particular stress conditions. Cells that had a high level of betacyanins show better resistance to stress in the cell culture, as well as, an overall different behavior including cell aggregation and alterations in nuclear size. Together the results shown here may provide new strategies to manipulate and mass-produce the pigments from Stenocereus queretaroensis in cell culture.